# Overcoming the Collaboration Barriers among Stakeholders in Urban Renewal Based on a Two-Mode Social Network Analysis

Lingyan Li [1,2] , Jiaxin Zhu [1,2,*], Mimi Duan [1,2], Pingbo Li [1,2] and Xiaotong Guo [1,2]

1    School of Management, Xi'an University of Architecture and Technology, Xi'an 710055, China
2    Key Research Base for Joint Construction and Sharing of Shaanxi Human Settlements and Good Life in the New Era, School of Management, Xi'an University of Architecture and Technology, Xi'an 710055, China
*    Correspondence: zhujiaxin@xauat.edu.cn

**Abstract:** The relationship among stakeholders is complicated and full of collaboration barriers, which makes urban renewal an intersection of various contradictions. However, the existing literature considers the barriers to urban renewal independent of stakeholders, and the interaction between multiple stakeholders and barriers to collaboration has been ignored. Therefore, this study uses a literature review and expert interviews to identify stakeholders and their collaboration barriers in the process of urban renewal. Based on the results of expert questionnaires, a two-mode network model of stakeholder–collaboration barrier is constructed to clarify the complex interaction and reveal the power and status of stakeholders in a network relationship. The study found that each barrier was associated with at least three stakeholders, indicating the necessity of stakeholders to establish partnerships. Further analysis shows that the government, local and other administrative organizations, consulting parties, and developers are the most influential stakeholders. The vague boundary of property rights, lack of expert advice and expertise, and different stakeholder awareness were identified as key barriers affecting sustainable collaboration. Finally, this study proposes and validates five strategies to promote collaboration among stakeholders. This study helps practitioners identify the priority problems to be solved under limited resources and provides effective measures to promote stakeholder collaboration.

**Keywords:** urban renewal; stakeholders; collaboration barriers; two-mode social network analysis

## 1. Introduction

Since the reform and opening up in the 1970s, China's economy and technology have continued to advance, and the urbanization process has developed by leaps and bounds, which has been accompanied by the aggregation and expansion of population and industries. The continuous expansion of the urban scale has caused land resource constraints in urban development. On this basis, urban construction has gradually changed from incremental expansion to potential tapping of stock. To achieve sustainable urban development and promote spatial structure optimization, industrial structure adjustment, and an excellent living environment [1], urban renewal has become a main urban construction improvement path.

Urban renewal is a process of the comprehensive action of various factors of the city, covering material, economic, cultural, ecological and other aspects. It is not only a unilateral building restoration or reconstruction but also involves the improvement of urban appearance, the reuse of land resources, the inheritance of urban history and culture, and the maintenance of social relations [2,3]. However, in view of the current actual situation, the implementation of urban renewal projects is very difficult, and most of them have not achieved the expected effect [4]. The reason is that urban renewal is not a single interest activity that can be completed by a single subject. It covers many aspects, such as project planning, project implementation and operation management, and it requires

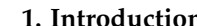

the collabotation of multiple subjects. Presently, problems, such as the different roles and goals of stakeholders [5], great differences in power, and difficulty in the unification of interest distribution [6] lead to complex relationships among stakeholders, great difficulty in coordination, and increased project implementation complexity. Concurrently, a strong distrust of social subjects exists, which leads to a series of problems, such as social fairness and conflict.

Considering the complexity of conflicts of interest and contradictions of stakeholders, the comprehensive benefits of urban renewal have been seriously affected. Some scholars have conducted evolutionary game analysis of urban renewal based on the perspective of core stakeholders to improve collaborative relationships [7]. However, although research focused on core stakeholder groups is beneficial for efficiently describing contradictions in the renewal, it ignores the impact of other groups on comprehensive benefits in the urban renewal process. Existing research focuses on the dilemmas faced by urban renewal [8], but it rarely involves the excavation of barriers that affect the complex collaborative relationship between stakeholders. It is noteworthy that the network governance of stakeholders in urban renewal has also become a research hot spot [9,10]. However, the single dimension of stakeholder research does not explore the collaborative relationship, and a lack of an analytical framework combining collaboration barriers with stakeholders to explore the interaction between stakeholders and collaboration barriers still exists.

In addition, in the existing research, it is still unclear how to establish collaboration among urban renewal stakeholders. Liu et al. believed that power represents the ability of stakeholders to overcome problems [11]. Different stakeholders have different powers and interests, which impact their behavior and final results [12]. To promote effective collaboration among urban renewal stakeholders, it is necessary to clarify their power to solve barriers. Therefore, this study will: (1) identify stakeholders in the process of urban renewal, explore the barriers that affect the collaboration of stakeholders, and surpass previous research limitations based on the perspective of core stakeholders and broad barriers; (2) establish the stakeholder–collaboration barrier two-mode social network of urban renewal, use quantitative methods to clarify the complex interaction between stakeholders and collaboration barriers, and reveal the power and status of stakeholders in the network relationship; (3) according to the core stakeholders and key collaboration barriers, propose targeted strategies to promote stakeholder collaboration and provide theoretical support and methodological guidance for stakeholders.

This study has theoretical and practical significance. Theoretically, it provides a new research perspective for analyzing the contradiction that urban renewal improves overall social benefits but leads to many social problems. It deeply analyzes the interactive relationship between stakeholders and collaboration barriers from a humanistic perspective, and it reveals the collaborative network structure of urban renewal stakeholders, which is conducive to accurately grasping the role and status of stakeholders. Practically, analyzing stakeholders' ability to overcome barriers to urban renewal provides strategic guidance for policy makers in urban renewal governance, which has important application value for the improvement of comprehensive benefits of urban renewal projects.

## 2. Literature Review

### 2.1. Stakeholders in Urban Renewal

Stakeholders are considered to be "any individual or group that can influence or be influenced by the achievement of an organization's objectives." [13,14] Stakeholders' perceptions, levels of knowledge, lack of commitment, and ineffective measures impact urban development projects [15]. In terms of stakeholder characteristics, power and interest are two important characteristics of stakeholders. The stakeholders' power can influence the scheme or benefit to a certain extent, whereas interest groups pay more attention to implementing the scheme or the final benefit [16,17]. Stakeholders with advantages in power can make great contributions to the project results or the subjects affected by them, and they will play a decisive role in the project [18]. In urban renewal, stakeholders

refer to groups that have vested interests in the results of urban renewal projects directly or indirectly. Stakeholders' power is analyzed to ensure effective collaboration between interest groups to achieve the desired objectives.

Collaboration provides more opportunities for stakeholders to participate [19]. The more frequently they communicate, the more conducive to the realization of project objectives [20]. In many studies, the collaboration of stakeholders is considered to be one of the most effective factors for the better implementation of urban renewal projects. Zhou et al. believe that the establishment of partnerships is the key to the implementation of urban renewal [21]. Soma et al. also believe that stakeholder interactions promote the transition of urban sustainability [22]. Zhang et al. indicate that improving multi-agent collaboration will contribute to successful urban renewal projects implementation [9]. Jutte et al. emphasize that stakeholders have different but complementary skills and expertise. Through cooperation, they can elaborate transformation plans and increase the possibility of seeking funds and establishing residents' relations [23].

Some scholars have paid attention to the difficulties of urban renewal. Huang et al. revealed the problems of sustainable renewal in terms of social, economic, and environmental aspects, land use forms, building conditions, and facilities [24]. Zhu et al. explored the barriers to sustainable renewal in communities and found that a lack of effective policy support was the root cause of unsustainable renewal in communities [25]. Many scholars have actively explored the network governance of urban renewal stakeholders. Guo et al. from the perspective of government reform and market-oriented management policies pointed out that the government is facing management difficulties due to insufficient understanding of owners' transformation intentions [26]. Bacigalupe et al. showed that the distrust of stakeholders would have a negative impact on the renewal project [27]. In the decision-making process of urban renewal, stakeholders' characteristics and interactive networks are highly complex [10].

However, the current theoretical research and practical experience of urban renewal focus on the exploration of the core interest groups and the broad difficulties, while the analysis of stakeholders in the entire process and barriers affecting stakeholder collaboration lacks a systematic excavation. Considering the intense conflicts between multiple stakeholders in urban renewal, this research focuses on the interaction between stakeholders and collaboration barriers to further analyze the status of stakeholders in the entire relationship network.

### 2.2. Two-Mode SNA

The study of social networks originating from the field of sociology reveals the interaction between individuals and the structure of interaction in society [28]. Individuals (actors) within the social network are called nodes, and the relationships between individuals are referred to as edges of the network. The strength of edges can represent their intimacy, contact time, and emotional intensity. Based on empirical data, SNA analyzes the specific structural relationships formed between social network subjects through computer technology [29] and then studies the interaction between these relationships and individual behavior.

Because this phenomenon, which is difficult to quantify, can be measured visually, many studies combine stakeholder analysis with SNA [30,31] to understand the complex formal and informal relationships between many stakeholders [32]. Contrary to traditional networks that only consider one node type, two-mode SNA studies the relationship between two different nodes. This link exists only between different node types and is used to evaluate the relationship between individuals and their attributes [33].

Some studies have added different attributes from the stakeholder perspective to evaluate the interaction between stakeholders. For example, Yu et al. studied the social risk management of stakeholders in the housing demolition stage of urban renewal projects [34]. Li et al. analyzed the social risks faced by stakeholders in implementing infrastructure projects using the two-mode social network method [35], and Xiang et al. explored the key factors of stakeholders in promoting an age-friendly community and developed a

preliminary framework [36]. Similarly, mapping stakeholders to the implementation barriers of urban renewal and optimizing the defects of systematic reflection in traditional research methods can help accurately grasp the interaction between stakeholders and provide a different perspective for the study of network relations.

## 3. Methods

This study aimed to identify the complex relationship between stakeholders and collaboration barriers in the urban renewal process through a literature review, expert interviews, expert questionnaires, and SNA. The specific steps of this study are shown in Figure 1.

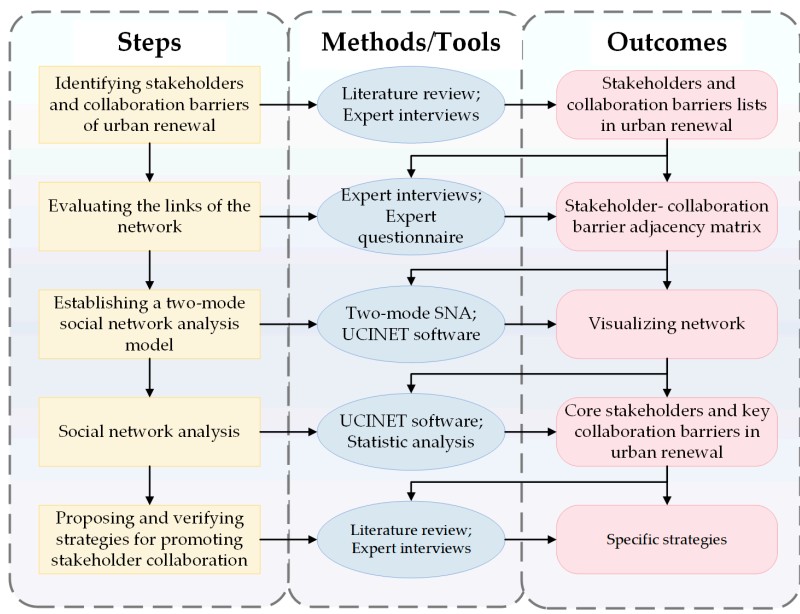

**Figure 1.** Research steps, methods, and results.

### 3.1. Step 1: Identifying Network Nodes

Nodes and links are important factors that constitute relational networks; therefore, identifying network nodes is the basis for building two-mode social networks. In this study, network nodes are the stakeholders of urban renewal and barriers affecting the collaboration of various participants. To identify these two types of nodes, a literature review and expert interviews were conducted. First, the previous research results were summarized through a literature review. In the preliminary list, there were 11 types of stakeholders and 18 barriers, which provided a basis for the identification of stakeholders and collaboration barriers [37].

Then, the expert interviews are used to verify the reliability of the node list, and the factors identified in the literature are revised to obtain a collection of urban renewal stakeholders and collaboration barriers that are accurately defined and easy to understand. The selection of interview experts needs to involve nine types of stakeholder representatives (except the owners and the public) and have more than five years of project managers or scientific research experience in the field of urban renewal. At first, two experts from academia and one general manager of the real estate company recommended other professionals in the field of urban renewal and then invited these experts to participate in the survey. Finally, ten experts from mainland China accepted the interview invitation. Table 1 below lists the experts interviewed.

**Table 1.** Interview experts for background information.

| Experts Number | Company Characteristics | Position | Years of Working |
|---|---|---|---|
| Experts 1 | Bank | Branch president | 25 |
| Experts 2 | Street office, community committee | Section chief of housing expropriation service center | 11 |
| Experts 3 | Real estate development company | General manager | 11 |
| Experts 4 | Expert scholar | Associate professor | 5 |
| Experts 5 | Real estate appraisal agency | Deputy general manager of real estate consulting company | 15 |
| Experts 6 | The media | Deputy editor-in-chief | 18 |
| Experts 7 | The government | Deputy director of Urban Planning Bureau | 30 |
| Experts 8 | Real estate appraisal agency | Real estate appraiser | 5 |
| Experts 9 | Design company | The engineer | 5 |
| Experts 10 | The contractor | Project manager of construction | 20 |

Experts needed to be clear about which stakeholder representative they were and answer the following questions: (1) Who are the stakeholders in urban renewal? Include whether the literature review list has unrelated stakeholders and what other stakeholders are not mentioned. (2) Which factors affect collaboration among stakeholders in urban renewal? Include whether the literature review list contains unrelated barriers, and what other barriers are not mentioned. (3) What methods, technologies, and policies can be used to promote collaboration among stakeholders in urban renewal?

*3.2. Step 2: Evaluating Network Links*

After the network nodes were determined, an expert questionnaire was used to explore the connection between the two node types, to clarify the power of stakeholders in solving collaboration barriers. The questionnaire mainly targeted stakeholders with rich experience and knowledge of urban renewal, such as government employees, university experts and scholars, senior management personnel of companies, and staff of third-party organizations. To ensure the rationality of the sample data, it is necessary to ensure the integrity of the sample data and control the balance of questionnaire quantity for each stakeholder.

The questionnaire was divided into three sections (see Appendix A). The first section aimed to understand the basic information of the interviewed experts, including their unit, work, and years of working in urban renewal. The second section identified the relationship between stakeholders and collaboration barriers in urban renewal projects. It listed stakeholders and barriers affecting stakeholders' collaboration, and it asked interviewees to evaluate which stakeholders could solve specific collaboration barriers according to their experience. Finally, the relationship between stakeholders before and after implementing the strategies was identified (see Section 3.5).

The questionnaires were distributed both online and offline. From August to November 2021, a paper expert questionnaire survey was distributed at the "13th National Existing Building Reconstruction Conference", and field research was conducted in the government, street offices, communities, banks, and other departments. A total of 187 valid questionnaires were recovered. In addition, it was sent to the interviewees online through the professional questionnaire survey platform "questionnaire star". The survey scope should not be too small because the respondents to the questionnaire should be familiar with urban renewal projects. To expand the sample size, questionnaires were distributed using snowball sampling. Snowball sampling is a non-probabilistic sampling method that can increase sample diversity in hard-to-reach populations [38]. In this study, ten experts who participated in the interviews were invited to distribute questionnaires to their colleagues, and these new respondents were also invited to distribute questionnaires to their partners to meet the requirements of snowball sampling. Finally, 108 valid questionnaires were collected online.

A total of 362 experts took part in the survey, of which 295 were valid, with an effective rate of 81.5%. In the overall sample, the respondents covered all stakeholder categories, of which 90.85% had participated in urban renewal projects, ensuring the accuracy of survey data and the quality of questionnaire recovery to a certain extent. The respondents' backgrounds are listed in Table 2.

**Table 2.** Background of respondents.

| The Unit Distribution | | Education Level | |
|---|---|---|---|
| Government | 8.47% | Junior college or below | 4.75% |
| Local and other administrative organization | 15.25% | Undergraduate | 61.36% |
| Consulting party | 17.29% | Master | 26.78% |
| Developer | 12.54% | Doctor and above | 7.12% |
| Financial institution | 12.20% | **Work experience** | |
| NGOs | 3.05% | Less than 1 year | 14.92% |
| Media | 5.08% | 1–5 years | 29.83% |
| Designers | 11.86% | 6–10 years | 17.29% |
| Contractor | 10.17% | 11–15 years | 20% |
| Property management company | 2.71% | 16–20 years | 5.76% |
| Operating agency | 1.36% | Over 20 years | 12.2% |

*3.3. Step 3: Establishing a Two-Mode SNA Model*

The SNA can be described using the community graph and matrix algebra methods. The former studies the relationship and structure among members of a group [39], whereas the latter describes the multiple relationships among different groups. Urban renewal projects are in a complex social network environment involving multiple stakeholders, and this study aims to reveal the power status of stakeholders regarding collaboration barriers. Therefore, matrix algebra was selected to express the relationship between the nodes. The matrix method used to describe the two-mode social network includes a set of stakeholders (X), a group of barriers (Y), and the relationship between two types of nodes ($a_{ij}$), where $X_i$ represents any one of the stakeholders, $Y_i$ represents any of the barriers, and $a_{ij}$ represents whether a stakeholder can solve a barrier. The definitions are:

$$a_{ij} = \begin{cases} 1, & \text{if } X_i \text{ is able to solve } Y_i \\ 0, & \text{otherwise} \end{cases}$$

According to the collected data, the data were sorted and summarized to obtain the stakeholder–collaboration barrier adjacency matrix, and then, UCINET software was used to present the two-mode SNA model of the stakeholder–collaboration barrier.

*3.4. Step 4: Social Network Analysis*

The SNA can explain the position, rights, and influence of each node in the relationship network as well as the influence and constraint of one party on another party. Centrality is an important method for analyzing network characteristics [24]. To further explore the influence of stakeholders on barriers, we chose degree centrality, eigenvector centrality, and betweenness centrality to measure and analyze network centrality.

In addition to analyzing the centrality of a network, network structure is also a method for evaluating the importance of nodes. Here, we focus on the core–periphery structure, which is composed of two parts. One part constitutes the core of dense connections, which is well connected with the peripheral nodes, and the other constitutes the periphery with sparse connection and edge distribution [40].

*3.5. Step 5: Proposing and Validating Strategies*

Through an in-depth understanding of the meaning of the two types of node links, based on the results of a two-mode SNA, combined with a literature review and expert interviews, we propose strategies to overcome the collaboration barriers of stakeholders. In the questionnaire distribution, following the formulation of the strategy, the validity of the strategy is verified in the third section, "the identification of the relationship network between stakeholders before and after the implementation of strategies". Respondents were asked to quantify their relationship with other stakeholders from their perspective, using a five-level scale to measure the collaboration between stakeholders, in which "1" and "5" indicate very low and very close relationship, respectively [41].

## 4. Results

### 4.1. Identification of Stakeholders and Collaboration Barriers

In the process of stakeholder identification, this study extracted 11 urban renewal stakeholders through academic literature, and the follow-up expert interviews suggested adding two more stakeholders. Based on the above analysis, 13 stakeholders involved in the urban renewal project were identified, and the list of stakeholders is shown in Table 3. The entire process of urban renewal and the corresponding stakeholders, according to the author's knowledge and practical experience, is shown in Figure 2.

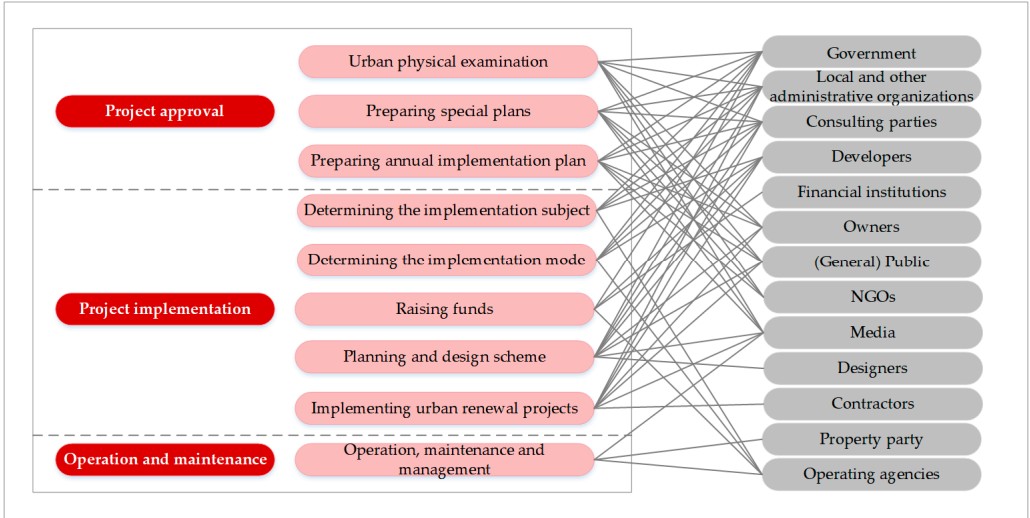

**Figure 2.** The entire process of urban renewal and corresponding stakeholders (by authors).

In addition, 18 barriers were initially summarized through a literature review to identify the barriers affecting stakeholder collaboration in urban renewal projects. In subsequent expert interviews, the experts' opinions were strongly consistent with the results of the literature review, and they recognized the 18 barriers initially identified. Specific barriers are presented in Table 4.

### 4.2. Stakeholder–Collaboration Barrier Adjacency Matrix

After questionnaires were collected, the SNA tool UCINET6.0 was used to conduct a consistency test on the questionnaire results to determine whether the respondents' cognition of the relationship between stakeholders and collaboration barriers was consistent and whether the data were reliable. The results are shown in Table 5. As shown in the results, the ratio between the largest eigenvalue and the second-largest eigenvalue was 14.601. When this index is greater than 3 in statistics, the data are considered consistent, which means that the 295 relational data have good stability.

**Table 3.** The stakeholders of urban renewal identified in previous literature and expert interviews.

| Stakeholder | Description | Key References |
|---|---|---|
| Government (S1) | Development and Reform Commission<br>Urban Planning Bureau<br>Land, Resources, and Housing Bureau<br>Urban and Rural Development Commission<br>Land and Resources Bureau | [10,42] |
| Local and other administrative organizations (S2) | Housing Administration Bureau<br>Street Office<br>Community Committee | [43–45] |
| Consulting parties (S3) | Experts and Scholars<br>Real Estate Evaluation Agencies<br>Consulting Companies Involved in Urban Renewal<br>And Urban Planning<br>Third-Party Companies Engaged in Smart<br>City/Smart Community/Smart Interconnection<br>Big Data Technology Service Companies | [10,46] |
| Developers (S4) | Real Estate Development Company | [5,6] |
| Financial institutions (S5) | Bank | [10,47] |
| Owners (S6) | Affected Residents | [6,48] |
| (General) Public (S7) | | [41,49] |
| Non-governmental organizations (NGOs) (S8) | Trade Associations<br>Associations of Entrepreneurs<br>Associations of Self-employed Workers | [46,50,51] |
| Media (S9) | TV<br>Radio<br>Internet Media | [50,52] |
| Designers (S10) | Urban Design Practitioners<br>Architects<br>Planners<br>Gardeners | [44,51,52] |
| Contractors (S11) | The Construction Companies | [44,53] |
| Property party (S12) | Property Management Companies | From expert interviews |
| Operating agencies (S13) | Relevant Business Operators | From expert interviews |

In this study, the standard deviation of network density was used as the segmentation value to convert the collected data into binary data. If it is greater than the standard deviation, there is a connection between stakeholders and collaboration barriers; then, it is represented by "1" in the table. Otherwise, it will be represented by "0". The results are presented in Table 6.

Regarding the influence of various stakeholder groups, Table 6 shows that among all stakeholder groups, S1 (government) can solve the most barriers (18 barriers), followed by S2 (local and other administrative organizations), S3 (consulting parties), and S4 (developers). B4 (vague boundary of property rights), B9 (lack of expert advice and expertise), B15 (different awareness of stakeholders), B16 (lack of participation willingness) and B17 (lack of participation policies) require the most stakeholders to overcome the barriers.

*4.3. Visualizing Stakeholder–Collaboration Barrier Network*

We imported the stakeholder–collaboration barrier adjacency matrix data into Net-Draw software from Analytic Technologies to realize visualization, as shown in Figure 3. The network was composed of 13 stakeholders and 18 impact problem nodes connected by 99 links. Red circular nodes represent stakeholders, blue square nodes represent collaboration barriers of urban renewal stakeholders, and lines represent the relationship between stakeholders and collaboration barriers.

**Table 4.** The barriers to stakeholder collaboration of urban renewal identified in previous literature and expert interviews.

| Collaboration Barrier | Description | Key References |
|---|---|---|
| Inadequate laws and regulations (B1) | There is no law on the theme of urban renewal in China, and only some cities have promulgated local regulations. | [10,54] |
| Frequent policy adjustments (B2) | Repeated policy changes affect the implementation of urban renewal projects. | [55] |
| Imperfect policy system (B3) | Many cities have yet to form a policy document system including special policies, technical standards and operational guidelines. | [56] |
| Vague boundary of property rights (B4) | The ownership of property is complicated and the ownership relationship is fuzzy, which easily leads to interest disputes. | [6,45] |
| Imperfect decision-making system (B5) | Strict separation of decision-making stage, lack of project planning guidance, lack of cross-departmental cooperation mechanism | [54] |
| Imperfect accountability mechanism (B6) | The imperfect accountability system affects the public's supervision and restriction of rights. | [57] |
| Unequal and opaque information (B7) | Information is asymmetric among stakeholders, and public participation cannot rely on effective and transparent information communication. | [25,58] |
| Complex coordination procedure (B8) | The process is complicated and lacks cross-field cooperation mechanism, which leads to the inability of effective coordination among departments. | [54] |
| Lack of expert advice and expertise (B9) | The participation channels of experts and scholars are not smooth, and cognition and judgment do not play their due roles in key decisions. | [57,59] |
| Imperfect (operational) management system (B10) | Most of the projects lack of industry authority, property management and other management organizations, resulting in the follow-up management work being difficult to continue. | [60] |
| Inadequate supervision (B11) | Lack of supervision subject and the imperfect supervision system, easy to free-ride, the performance of duties is not in place and so on. | [25,56] |
| Unbalanced distribution of benefits (B12) | It is difficult for stakeholders to reach an agreement on urban renewal compensation and social benefit distribution. | [45,61] |
| Single financing model (B13) | Single investment entity, limited funds, use market mechanism to introduce social capital insufficiency | [57] |
| Inefficient investment of capital (B14) | The short-term economic benefits of some renewal projects are low, and the income and expenditure of funds are difficult to balance. | [8] |
| Different awareness of stakeholders (B15) | There is a strong distrust of one group toward the other in moving cities on a more sustainable path. | [10,42] |
| Lack of participation willingness (B16) | The public tends to rely on the government to make decisions in China. They usually pay more attention to their own interests than to the public interest. | [48,58] |
| Lack of participation policies (B17) | There are still no policies for the public on how to express their demands and realize the right to know, participation and supervision of management. | [42,58] |
| Problems of social equity (B18) | On the aspects of law and policy, it neglects the protection of social fairness and the interests of vulnerable groups and affects the system fairness and democracy. | [25,56,62] |

**Table 5.** Questionnaire consistency analysis test results.

| No. of Negative Competencies | 3 |
| --- | --- |
| Largest eigenvalue | 141.754 |
| 2nd largest eigenvalue | 9.708 |
| Ratio of largest to next | 14.601 |

**Table 6.** Stakeholder–collaboration barrier adjacency matrix.

| | B1 | B2 | B3 | B4 | B5 | B6 | B7 | B8 | B9 | B10 | B11 | B12 | B13 | B14 | B15 | B16 | B17 | B18 | SUM |
| --- | --- | --- | --- | --- | --- | --- | --- | --- | --- | --- | --- | --- | --- | --- | --- | --- | --- | --- | --- |
| S1 | 1 | 1 | 1 | 1 | 1 | 1 | 1 | 1 | 1 | 1 | 1 | 1 | 1 | 1 | 1 | 1 | 1 | 1 | 18 |
| S2 | 1 | 1 | 1 | 1 | 1 | 1 | 1 | 1 | 1 | 1 | 1 | 1 | 0 | 0 | 1 | 1 | 1 | 1 | 16 |
| S3 | 1 | 1 | 1 | 1 | 1 | 1 | 1 | 1 | 1 | 1 | 0 | 0 | 0 | 0 | 1 | 1 | 1 | 0 | 13 |
| S4 | 1 | 0 | 1 | 1 | 1 | 1 | 1 | 1 | 1 | 1 | 0 | 1 | 1 | 1 | 1 | 0 | 0 | 0 | 13 |
| S5 | 0 | 0 | 0 | 0 | 0 | 0 | 0 | 0 | 0 | 0 | 0 | 0 | 1 | 1 | 0 | 0 | 0 | 0 | 2 |
| S6 | 0 | 0 | 0 | 1 | 0 | 1 | 0 | 0 | 0 | 0 | 1 | 1 | 0 | 0 | 1 | 0 | 1 | 0 | 6 |
| S7 | 0 | 0 | 0 | 1 | 0 | 0 | 1 | 0 | 0 | 0 | 1 | 0 | 0 | 0 | 1 | 1 | 1 | 1 | 7 |
| S8 | 0 | 0 | 0 | 0 | 0 | 0 | 0 | 0 | 1 | 0 | 1 | 0 | 0 | 0 | 0 | 1 | 1 | 0 | 4 |
| S9 | 1 | 1 | 1 | 1 | 0 | 0 | 1 | 0 | 0 | 0 | 1 | 0 | 0 | 0 | 1 | 1 | 1 | 1 | 10 |
| S10 | 0 | 0 | 0 | 0 | 1 | 0 | 0 | 0 | 1 | 0 | 0 | 0 | 0 | 0 | 0 | 0 | 0 | 0 | 2 |
| S11 | 0 | 0 | 0 | 0 | 0 | 0 | 0 | 0 | 0 | 0 | 0 | 0 | 0 | 0 | 0 | 0 | 0 | 0 | 0 |
| S12 | 0 | 0 | 0 | 0 | 0 | 0 | 0 | 1 | 0 | 1 | 0 | 0 | 0 | 0 | 0 | 0 | 0 | 0 | 2 |
| S13 | 0 | 0 | 0 | 0 | 1 | 0 | 0 | 1 | 1 | 1 | 0 | 0 | 1 | 0 | 0 | 1 | 0 | 0 | 6 |
| SUM | 5 | 4 | 5 | 7 | 6 | 5 | 6 | 6 | 7 | 6 | 6 | 4 | 4 | 3 | 7 | 7 | 7 | 4 | |

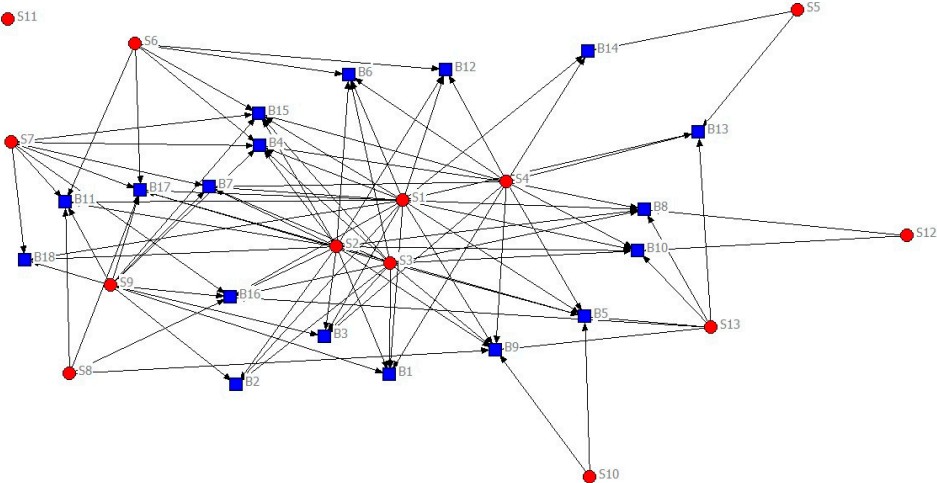

**Figure 3.** Visualization of the stakeholder–collaboration barrier network.

By transforming the stakeholder–collaboration barrier matrix into the stakeholder–stakeholder matrix and the collaboration barrier–collaboration barrier matrix, the visualization results are derived, as shown in Figures 4 and 5. Specifically, Figure 4 shows the network of stakeholder relationships, and the values on the lines represent the number of barriers that stakeholder a and stakeholder b can solve simultaneously. Among all stakeholder groups, S1 (government) and S2 (local and other administrative organizations) have the highest similarity and the ability to affect 16 barriers simultaneously, showing that both parties are more likely to establish collaboration and take concerted actions based on complementary advantages and shared responsibilities. The S3 (consulting parties), S4 (developers), and S9 (media) are also closely associated with these barriers. In contrast, many nodes are not connected to S5 (financial institutions) and S12 (property management companies), which means that it is difficult to take concerted action on the same barrier to promote the implementation of urban renewal.

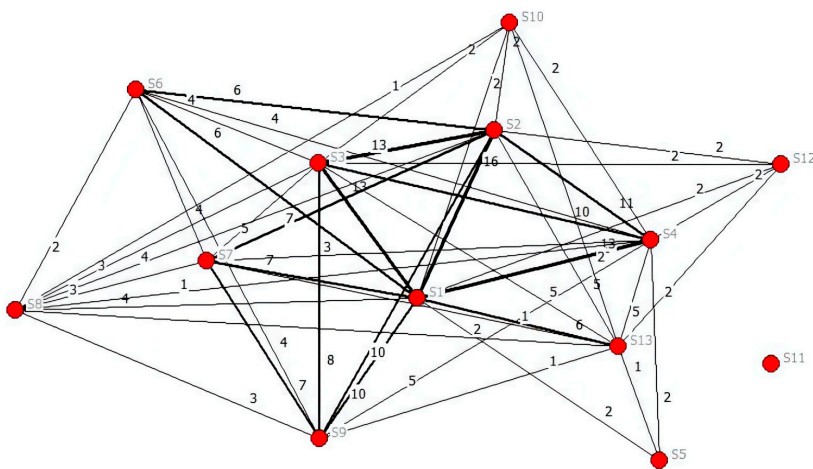

**Figure 4.** Visualization of the stakeholder–stakeholder network.

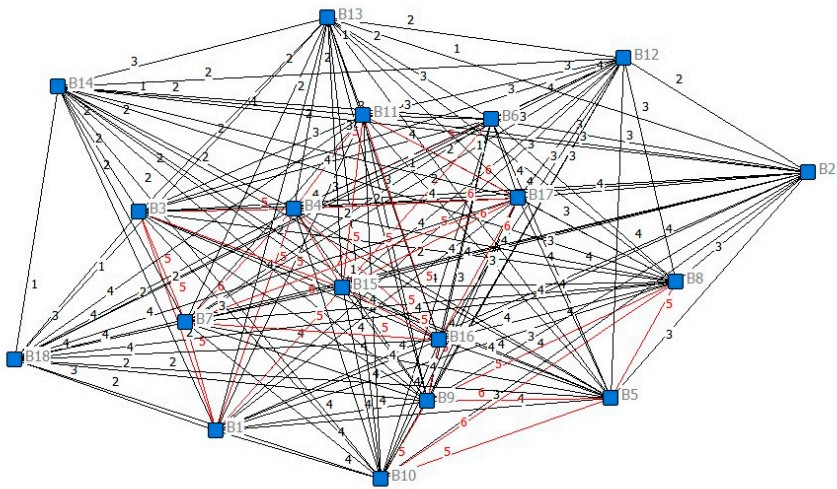

**Figure 5.** Visualization of the collaboration barrier–collaboration barrier network.

Figure 5 shows the network of collaboration barriers. The values on the lines represent the number of barriers a and b that can be solved by stakeholders simultaneously, which can test the resource similarity of barriers [63]. The weight of the red line is not less than 5. The values greater than one in the matrix indicate that each pair of barriers can be addressed by at least one stakeholder group. The matrix shows that barriers B4 (vague boundary of property rights) and B15 (different awareness of stakeholders) can be affected by the largest number of stakeholder groups simultaneously; thus, they have a high resource similarity to barriers and require similar stakeholder groups to take action. By contrast, the values associated with disorder B14 (inefficient investment of capital) were mostly 1.

*4.4. Centrality and Core–Periphery Structure*

The relationship between nodes was analyzed using three index measures: degree, eigenvector, and betweenness centrality. The top three nodes were selected as the core barriers and stakeholders [64]. If the value below the top three nodes was the same as that of the third, they were also considered core node elements.

The results of Figure 6 show that S1 (government) has the largest degree centrality of 1, which is followed by S2 (local and other administrative organizations), S3 (consulting parties) and S4 (developers) in the third position together. This indicates that these four stakeholders occupy an important position in the network, have the most direct influence, and have the power to establish partnerships with broader stakeholders to address different barriers. In terms of eigenvector centrality, S1 (government), S12 (property party), and S2

(local and other administrative organizations) have the highest scores, indicating that these three stakeholders can solve more critical barrier elements. Similarly, S1 (government), S2 (local and other administrative organizations), and S4 (developers) have the highest betweenness centrality scores, constitute the core of the network, play an important role in the network's connection, and exert strong influence and control on the barriers in the network. The ranking of betweenness centrality and degree centrality is basically the same, which is consistent with the findings of Xu et al. [63].

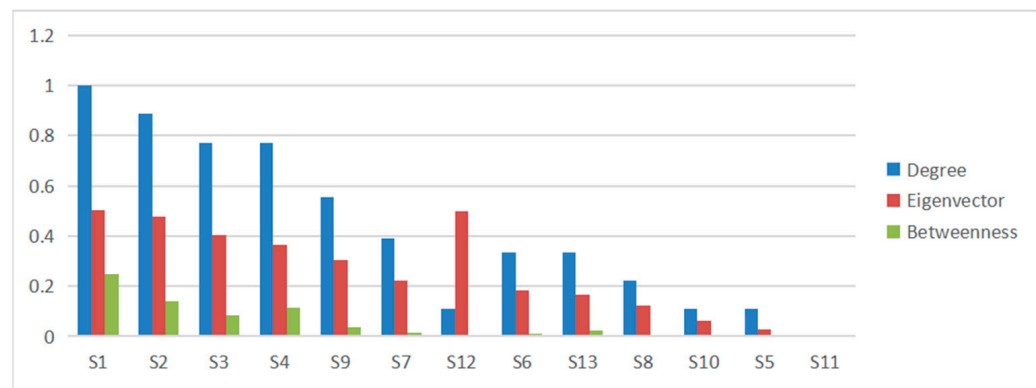

**Figure 6.** Centrality of stakeholder nodes.

As shown in Figure 7, B4 (vague boundary of property rights), B9 (lack of expert advice and expertise), B15 (different awareness of stakeholders), B16 (lack of participation willingness), and B17 (lack of participation policies) have the highest degree centrality scores. This shows that these five barriers impact other barriers and must be addressed by a wider range of stakeholders. B4 (vague boundary of property rights) and B15 (different awareness of stakeholders) have the highest eigenvector centrality scores, which are followed by B7 (unequal and opaque information). This shows that these barriers connect more stakeholders and are more likely to impact the network than other barriers. For example, if information inequality and opacity are not effectively addressed, it may lead to different stakeholder awareness and social equity issues. The three barriers with the highest betweenness centrality scores are B9 (lack of expert advice and expertise), B8 (complex coordination procedure), and B10 (imperfect (operational) management system). This means that they act as intermediaries and bridges with a great ability to alter or obstruct the information passing through them.

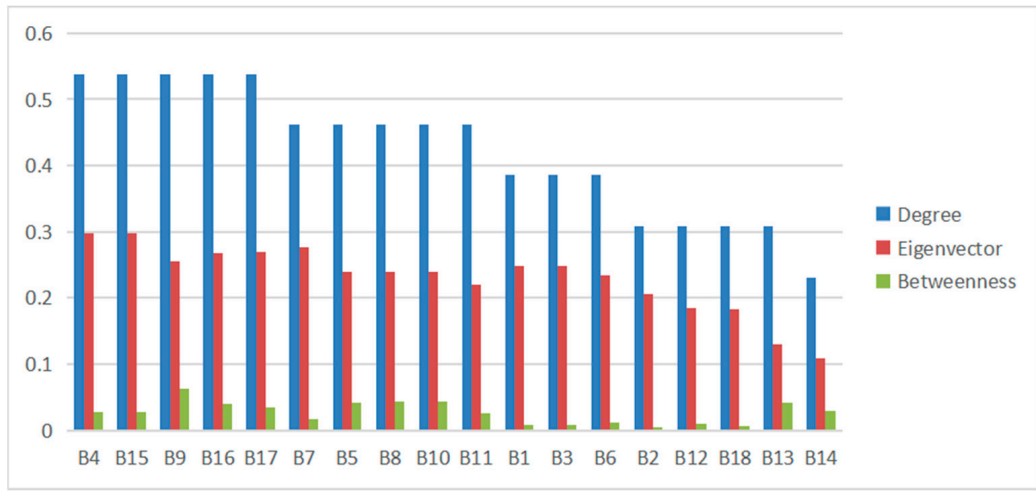

**Figure 7.** Centrality of collaboration barrier nodes.

The density matrix of the core–periphery analysis results is presented in Table 7. The fitness value is between 0 and 1. The larger the value is, the higher the fitness will be. Thus, the two types of nodes are closely related. A final fitness of 0.772 indicates that this network structure conforms to the ideal core–periphery structure. The density between the core stakeholders and the core barriers is 0.918, showing a strong relationship between stakeholders at the core of urban renewal and barriers. It can be considered that the stakeholder–collaboration barrier network presents a core–periphery structure with high core relationship density.

**Table 7.** Density matrix.

| | | Barrier | |
|---|---|---|---|
| | | Core | Periphery |
| **Stakeholder** | Core | 0.918 | 0.579 |
| | Periphery | 0.273 | 0.140 |
| | Final fitness: 0.772 | | |

As shown in Table 8, the core node identifies four stakeholders and 13 barriers in the upper-left corner. The four core stakeholders are S1 (government), S2 (local and other administrative organizations), S3 (consulting parties), and S4 (developers), accounting for about one-third of the total number of stakeholders. Except for barriers B14 (inefficient investment of capital), B2 (frequent policy adjustments), B12 (unbalanced distribution of benefits), B13 (single financing model), and B18 (problems of social equity), the remaining 13 barriers are in the core position, accounting for over two-thirds of the total barriers. Core stakeholders have more power and resources to take action against these core barriers, which is also the key to coordinating other stakeholders to establish collaborative relationships. Core stakeholders are likely to have close interactions and increased opportunities to exchange information, which is conducive to forming common values, attitudes, and interests in urban renewal [64]. In addition, if the core barrier can be solved, the transmission path from the core barrier to the peripheral barrier can be blocked, and the peripheral barrier can be well controlled.

**Table 8.** Core–periphery structure model of stakeholder–collaboration barrier network.

| | B1 | B11 | B3 | B4 | B5 | B6 | B7 | B8 | B9 | B10 | B15 | B16 | B17 | B14 | B2 | B12 | B13 | B18 |
|---|---|---|---|---|---|---|---|---|---|---|---|---|---|---|---|---|---|---|
| S1 | 1 | 1 | 1 | 1 | 1 | 1 | 1 | 1 | 1 | 1 | 1 | 1 | 1 | 1 | 1 | 1 | 1 | 1 |
| S2 | 1 | 1 | 1 | 1 | 1 | 1 | 1 | 1 | 1 | 1 | 1 | 1 | 1 | | 1 | 1 | | 1 |
| S3 | 1 | | 1 | 1 | 1 | 1 | 1 | 1 | 1 | 1 | 1 | 1 | 1 | 1 | | | | |
| S4 | 1 | | 1 | 1 | 1 | 1 | 1 | 1 | 1 | 1 | 1 | | | 1 | | 1 | 1 | |
| S5 | | | | | | | | | | | | | | 1 | | | 1 | |
| S6 | | 1 | | 1 | | 1 | | | | | 1 | | 1 | | | 1 | | |
| S7 | | 1 | | 1 | | | 1 | | | | 1 | 1 | 1 | | | | | 1 |
| S8 | | 1 | | | | | | | 1 | | | 1 | 1 | | | | | |
| S9 | 1 | 1 | 1 | 1 | | | 1 | | | | 1 | 1 | 1 | 1 | | | | 1 |
| S10 | | | | | 1 | | | | 1 | | | | | | | | | |
| S11 | | | | | | | | | | | | | | | | | | |
| S12 | | | | | | | | 1 | | 1 | | | | | | | | |
| S13 | | | | | 1 | | | 1 | 1 | 1 | 1 | | | | | | 1 | |

## 5. Discussion

### 5.1. The Power of Stakeholders in Urban Renewal

Table 6 shows the power distribution of stakeholders on barriers [63]. Specifically, stakeholders S1, S2, S3, S4, S9, and S13 have a strong influence on policy and management barriers (B1, B2, B3, B4, B5, B6, B8, B9, B10, and B11), while the four types of stakeholders S1, S4, S5, and S13 are more capable of solving financial problems (B12, B13, and B14). S1, S2, S7, and S9 have more power to solve public participation barriers (B17, B18). In

addition, stakeholders S3 and S4 had greater influence on communication barriers (B7, B15, and B16).

In addition, the results show that each barrier is associated with at least three or more stakeholders, indicating the importance of collaboration among stakeholders in achieving project objectives. Through the resource advantages of stakeholders, partners can be established to improve efficiency, solving the urban renewal barrier problem. This is consistent with the view of Wang et al. that inter-organizational linkages are very complex and require the interconnectedness and multi-faceted alignment of stakeholders [65]. Among these stakeholders, special attention should be paid to the role of S3 (consulting parties), which is often overlooked in urban renewal. However, S3 (consulting parties)'s professional skills not only serve as an important reference for S1 (government) and S2 (local and other administrative organizations) but also make it easier to design a dual scheme with both cultural protection and implementation. In addition, stakeholder S11 (contractors) in the adjacency matrix cannot resolve barriers in the list, because S11 (contractors) is better at solving technical problems in urban renewal projects, while the collaboration barrier in this study does not involve technical aspects, so the influence of S11 (contractors) is at the bottom.

Although S6 (owners) and S7 (public) are direct beneficiaries of urban renewal, the adjacency matrix results show that these two are not core stakeholders, which may be owing to the following reasons. On the one hand, this study mainly examines the power and status of stakeholders in solving barriers, while owners and the public do not have expertise and influence in overcoming barriers. On the other hand, owners and public individuals tend to have strong demands and can spread quickly, but it is always difficult to obtain an effective response from their demands. It may be that collective interests are difficult to be consistent, resulting in the low efficiency of their actions and high limitations.

It is worth noting that problems with communication and public participation can be addressed by a larger group of stakeholders than by other types of barriers. This means that overcoming these two types of barriers is complex, as more stakeholder groups are required to coordinate. This result is also in line with the opinion of Yu et al. that bottom–up efforts to promote urban renewal are still insufficient [66]. During the entire process, as the ultimate user, the owner lacks a voice channel, resulting in an extremely unbalanced role relationship. Therefore, it is particularly important to provide a legal platform for owners and the public to express their demands and to ensure the adoption and implementation of their suggestions.

### 5.2. Core Stakeholders and Key Collaboration Barriers in Urban Renewal

According to Figure 8a, the comprehensive centrality of stakeholders in the network shows that S1 (government), S2 (local and other administrative organizations), S3 (consulting parties), and S4 (developers) play an important role and have an important influence on other stakeholders. They can have a substantial impact on the exchange of information and resources in a network [67], are key to establishing synergies and influencing barriers, and are closely related to urban renewal projects. However, S5 (financial institutions), S8 (NGOs), and S11 (contractors) have poor centrality in the network and can be regarded as marginal stakeholders. They have weak influence on other stakeholders and are not directly related to urban renewal projects.

Similarly, an analysis of the centrality of the urban renewal barriers is shown in Figure 8b; B4 (vague boundary of property rights), B9 (lack of expert advice and expertise), and B15 (different awareness of stakeholders) are key elements that affect the synergistic effect of stakeholders in urban renewal, which needs the attention of stakeholders. In addition, the centralities of B16 (lack of participation willingness) and B17 (lack of participation policies) are also high; therefore, special attention should be paid to the governance of these elements. In contrast, marginal barriers, such as B2 (frequent policy adjustments), B12 (unbalanced distribution of benefits), B14 (inefficient investment of capital), and B18

(problems of social equity), are often caused by other barriers, which indirectly affect the establishment of collaborative relationships among stakeholders.

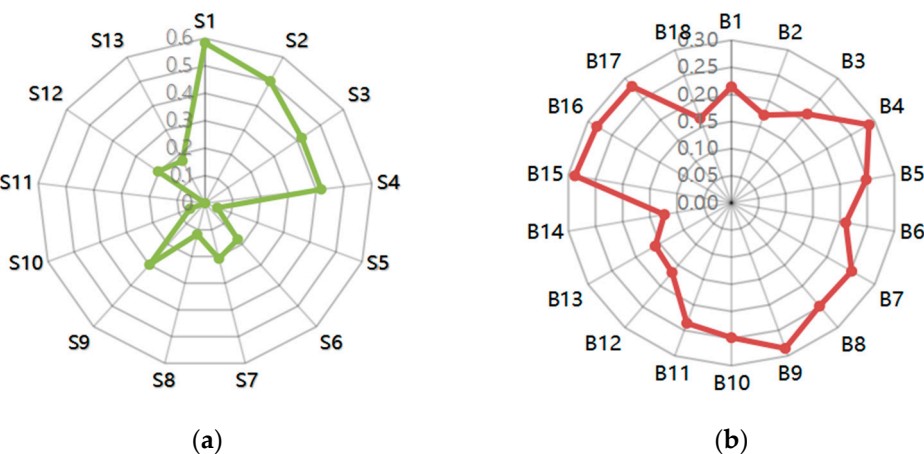

**Figure 8.** (**a**) Comprehensive centrality of stakeholders; (**b**) Comprehensive centrality of collaboration barriers.

### 5.3. Stakeholders' Collaboration Path in Urban Renewal

The core–periphery analysis results provide a reference for the collaboration path of stakeholders (see Figure 9). Core stakeholders play the role of key coordinators. They can bridge and transfer resources to the maximum extent, which is crucial to the overall success of urban renewal projects. That is, core stakeholders should establish close cooperation and use their own capabilities and resource advantages to coordinate conflicts and contradictions in the process. At the same time, due to the high centrality, core stakeholders should formulate management strategies, improve the interaction with other stakeholders, and give priority to mitigating key collaboration barriers. As a result, it can weaken the marginal barriers, promote the collaboration of all stakeholders, and facilitate the realization of the entire project objectives.

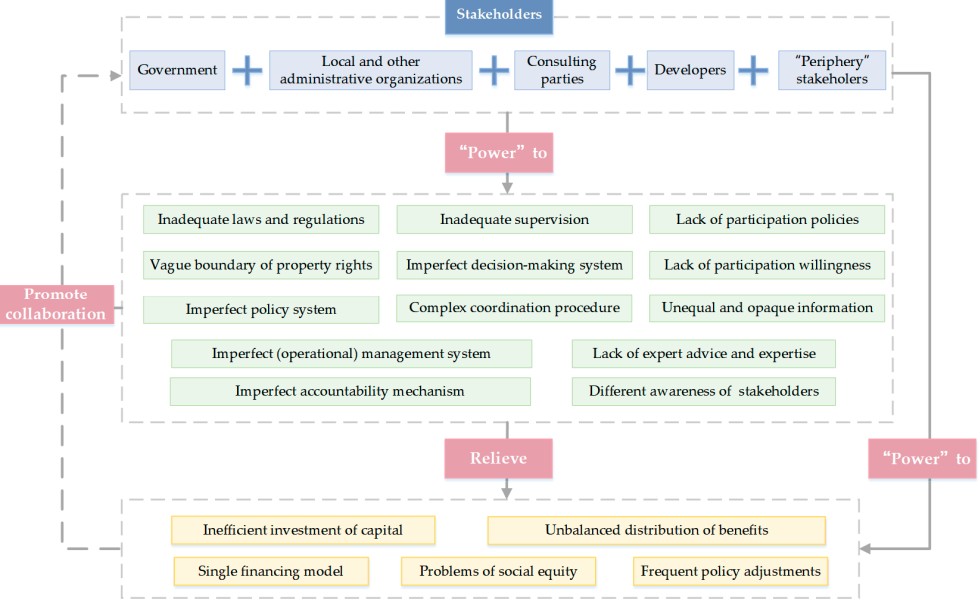

**Figure 9.** Stakeholders' collaboration path in urban renewal.

## 6. Strategies Proposal and Verification

### *6.1. Strategies to Promote Collaboration*

Through social network analysis, the core stakeholder of urban renewal, the key barriers affecting collaboration and the interaction between them are identified. Based on the author's knowledge and expert opinions, five strategies to promote collaboration are formulated from the perspective of core stakeholders. Figure 10 shows the effect of strategies. The square represents each strategy, the ellipse represents the key coordination factors of stakeholders, and the arrow represents the effect of strategies on key coordination factors: that is, these strategies can promote effective communication and mutual trust among stakeholders, and they can play a positive role in achieving the overall interests and common goals. At the same time, Strategy 3 and Strategy 5 can also ease the contradictions and conflicts among stakeholders.

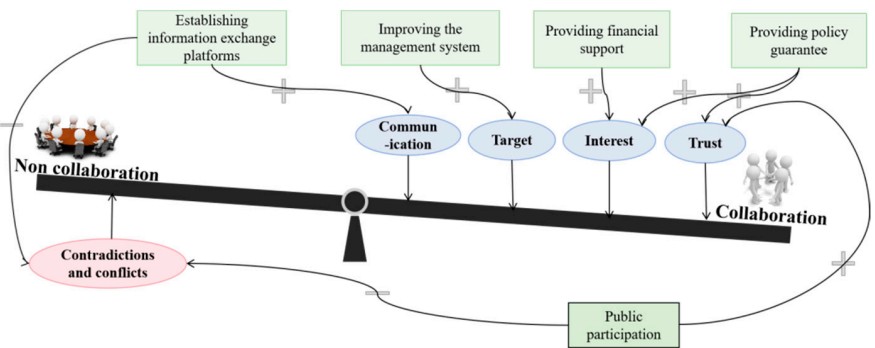

**Figure 10.** The effect of strategies (by authors).

### 6.1.1. Providing Policy Guarantee

Experts 5 and 7 believe that formulating detailed laws and regulations is the first step in urban renewal. Many problems exist in the practice of urban renewal and contradictions between existing planning laws and regulations; therefore, more comprehensive, systematic, and operational rules and regulations are required to connect them. By referring to the legislative experience of other mature regions in urban renewal, S1 (government) and S2 (local and other administrative organizations) also need to formulate a specialized "Urban renewal law" based on existing laws and regulations and in combination with existing local rules and regulations; this is necessary to clearly define the basic content, implementation procedures, operation procedures, demolition compensation, public interests, responsibilities and obligations of all stakeholders of urban renewal. They need to constantly make adjustments and improvements in the process of practice so that stakeholders can have laws to follow at all stages.

### 6.1.2. Improving the Management System

Experts 4 and 8 emphasize that a good management system is an important guarantee for urban renewal implementation. First, to achieve more effective participatory planning, a clear division of authority and better working mechanism are needed. S1 (government) and S2 (local and other administrative organizations) participate in policy formulation, planning approval, investment attraction, supervision, and management; they guide S4 (developers) to carry out development and construction, and they participate in the formulation of renewal plans; S3 (consulting parties) provides consulting services and encourages S6 (owners) to participate in the management of community public affairs. Second, we established an appropriate regulatory system to improve the accountability system. S1 (government) and S2 (local and other administrative organizations) should actively collaborate with relevant departments in multi-department supervision to safeguard the legitimate rights and interests of stakeholders. It is necessary to improve the accountability system, clarify accountability standards, and encourage S7 (public), S8 (NGOs), and S9 (media) to supervise and report them.

### 6.1.3. Establishing Information Exchange Platforms

Information sharing is the basis of sustainable urban renewal and effective stakeholder participation [6]. Key stakeholders should first establish alliances, integrate electronic information platforms, and share more diversified and accurate information (such as standards, policies, transformation plans, and schemes) to all stakeholders. They should pay attention to the information shared by stakeholders in real time to seek opportunities for collaboration among all stakeholders. The social network platform can be used as a medium to convey the parties' intentions, negotiations, and feedback throughout the project. S1 (government) and S2 (local and other administrative organizations) lead the establishment and implementation of interactive communication mechanisms (expert 4), and the electronic information platform of key stakeholders will integrate and coordinate the collected opinions, which will help improve information transparency and reach consensus among stakeholders.

### 6.1.4. Providing Financial Support

Experts 2, 5, and 8 emphasize the need for strong fiscal policies to support urban renewal projects and the need to find other financial resources to address B13 (Single financing model). Therefore, we should not only give full play to the guiding role of government financial funds but also introduce social investment through institutional innovation. For the former, S1 (government) can establish a special fund for urban renewal and reconstruction by drawing a proportion of a series of expenses such as land transfer income to support the renewal projects in S4 (developers) difficult or non-urban core areas. For the latter, S1 (government) can introduce social capital, such as establishing the S1 (government) and social capital collaboration model (PPP) [25] and reducing S1 (government) investment cost, to relieve the pressure of raising reconstruction funds. In addition, urban renewal involves the interests of many rights subjects, and no unified standard exists for their respective income distribution proportions. It is a huge workload to reach an agreement through consultations. It is urgent for S1 (government) to strengthen guidance and services, reasonably guide the income distribution expectations of renewal subjects and rights holders and safeguard the legitimate rights of S6 (owners) (expert 7).

### 6.1.5. Public Participation

Based on the visualization, it was observed that S6 (owners) and S7 (public) tended to be marginalized in the relationship network, and both were direct beneficiaries of the urban renewal project. Therefore, it is necessary to guarantee the rights of owners and the public (expert 8). This can be completed in the following ways: First, S1 (government) and S2 (local and other administrative organizations) should carry out publicity and related educational activities of urban renewal knowledge. Local newspapers, community announcements, TV, the Internet, seminars, and other forms can be used to inform owners and the public of the project situation in real time. Legal education should also be carried out to improve the quality of owners and the public. Second, they should use a variety of network platforms with the help of S8 (NGOs) and S9 (media) power to timely grasp the owners' renovation intentions. Third, S1 (government) and S2 (local and other administrative organizations) should add complaint channels for urban renewal projects and publicize rights protection methods and supervision channels. They should exercise the right of supervision through these channels when the interests of owners and the public are violated.

### 6.2. Verifying the Validity of Strategies

According to the collected 320 questionnaires (besides expert questionnaires, surveys between owners and the public were added), the relationship between stakeholders before and after the implementation of the strategies was identified to verify the validity of the five strategies. The results for the stakeholder relationship network are presented in Figure 11. The link indicates the relationship and the width indicates the closeness of the relationship between stakeholders. Compared with the original network, the link width of the new

network is generally increased, which means a higher degree of collaboration among stakeholders. The density of the new network is 4.421, which is an increase of 50.52% compared with 2.937 of the original network, showing increased collaboration between stakeholders, closer connections, and more convenient transmission of resource elements, and it also means that these five strategies can significantly ameliorate the collaboration barriers of urban renewal stakeholders and effectively improve the partnership.

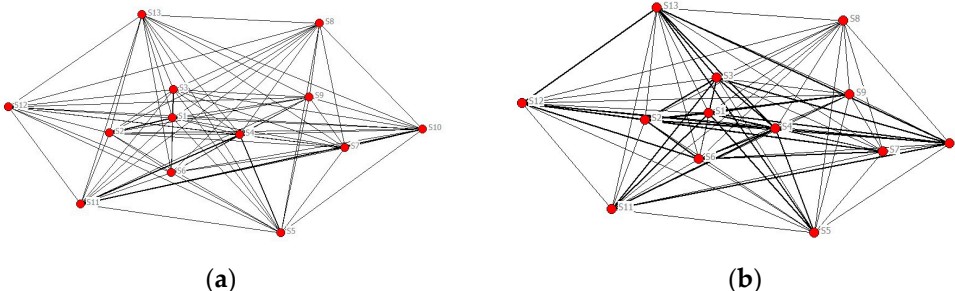

(**a**)                                                                                   (**b**)

**Figure 11.** Stakeholder relationship network before (**a**) and after (**b**) implementation of strategies.

## 7. Conclusions

This study provides a new research perspective on the collaboration among stakeholders in urban renewal. From the perspective of a two-mode social network, it discusses the influence of stakeholders on the barriers to collaboration in urban renewal, which is conducive to promoting the collaboration of stakeholders and the sustainability of urban renewal. After the literature review and expert interviews, a list of 13 stakeholders and 18 barriers to collaboration for urban renewal projects was drawn up. According to the results of the expert questionnaire survey, the correlation between stakeholders and barriers was confirmed, and the power status of stakeholders in different types of barriers and barriers was clarified. Each barrier is associated with at least three stakeholders, indicating that building partnerships is critical. Through a series of parameter analyses of network centrality, it was found that the government, local and other administrative organizations, consulting parties, and developers are the most influential stakeholders, and the vague boundary of property rights, lack of expert advice and expertise, and different awareness of stakeholders are key barriers to sustainable collaboration. The core–periphery results show that there are four stakeholders and 13 barriers at the core of the network, which also provides guidance for the establishment of an active stakeholder collaboration network. To overcome these barriers, policy, management, information, and economic and social measures have been proposed from the stakeholder perspective. The validity of these strategies was evaluated by calculating the densities of the initial and new stakeholder networks. The results show this strengthens the relationship between stakeholders.

This study explains the stakeholders in the entire process of urban renewal and the barriers to collaboration at the current stage, which have profound significance for stakeholders in improving the efficiency of urban renewal. Stakeholders have different advantages regarding different barriers, which provides guidance on which groups they should choose to work with and which barriers they should prioritize in the context of limited resources. This research highlights the importance of stakeholder collaboration in urban renewal from the perspective of multiple integration, while existing research focuses more on a single attribute, such as only studying stakeholders or barriers, and it pays insufficient attention to these interactions. In addition, when proposing strategies, existing studies ignore the correlation between stakeholders and barriers. Therefore, considering the correlation between stakeholders and collaboration barriers, this study proposes effective methods to promote stakeholder collaboration and smooth and efficient implementation of urban renewal to fill this research gap.

Although this study provides practical enlightenment for urban renewal, there are still some aspects that need to be improved. First, when discussing the relationship between stakeholders and barriers, an experience-based method is adopted, and the results of expert questionnaires are subjective and highly dependent on the experience level of experts. Further research can be conducted by combining subjective and objective methods, such as expert questionnaires and case analyses, to more accurate research results. Second, this study constructs a static network of urban renewal projects that lacks the network construction of the entire life cycle. However, different stages may lead to changes in the stakeholder power. In the future, differentiated analysis can be conducted based on the process of each stage of urban renewal. Third, the data in this study were from China, and the results can only reflect the status and resources of stakeholders in the process of urban renewal in China, which differs greatly from those in countries with more mature urban renewal development. Therefore, a comparative analysis can be conducted with these countries to provide experience in the process of urban renewal in China.

**Author Contributions:** Conceptualization, L.L. and J.Z.; formal analysis, J.Z. and P.L.; investigation, J.Z., M.D. and P.L.; methodology, L.L.; software, J.Z. and P.L.; supervision, L.L., M.D. and X.G.; validation, L.L., J.Z. and X.G.; visualization, J.Z. and M.D.; writing—original draft preparation, J.Z.; writing—review and editing, L.L. and J.Z.; funding acquisition, L.L. All authors have read and agreed to the published version of the manuscript.

**Funding:** This research was supported by the National Natural Science Foundation of China (Grant No. 71874136), the Humanities and Social Sciences Foundation, Ministry of Education of the People's Republic of China (Grant No. 19YJC630080), the Social Science Foundation of Shaanxi Province (Grant No. 2021ZD1015), the Education Department of Shaanxi Province (Grant No. 16JZ038) and the State Key Laboratory of Green Building in Western China (Grant No. LSZZ202202).

**Data Availability Statement:** Not applicable.

**Conflicts of Interest:** The authors declare no conflict of interest.

## Appendix A. The Sample of Expert Questionnaire

I. Basic information

(1)  Nature of your organization:
     ☐ Government ☐ Local and other administrative organization ☐ Consulting party ☐ Developer ☐ Financial institution ☐ NGOs ☐ Media ☐ Designer ☐ Contractor ☐ Property management company ☐ Operating agency ☐ Others
(2)  Your education background:
     ☐ Doctor ☐ Master ☐ Undergraduate ☐ Junior college ☐ High school ☐ Others
(3)  Your work experience:
     ☐ Less than 1 year ☐ 1–5 years ☐ 6–10 years ☐ 11–15 years ☐ 16–20 years ☐ Over 20 years
(4)  Have you ever participated in projects/studies related to urban renewal:
     ☐ No ☐ 1–3 ☐ More than 3

II. Identification of the relationship between stakeholders and collaboration barriers in urban renewal projects

In each blank in the matrix, you need to fill in numbers indicating the relationship between stakeholders and collaboration barriers in urban renewal.

"1" represents the stakeholder can solve the barrier;

"0" represents the stakeholder cannot solve the barrier;

III. Identification of the relationship between stakeholders in urban renewal projects

Please judge the close relationship between different stakeholders in the urban renewal project in the following matrix (before and after the implementation of strategies). The scoring criteria are as follows:

1 = very not close; 2 = not close; 3 = generally close; 4 = close; 5 = very close.

**Table A1.** The relationship between stakeholders and collaboration barriers in urban renewal projects.

| Stakeholders | Collaboration Barriers | | | | | | | | | | | | | | | | | |
|---|---|---|---|---|---|---|---|---|---|---|---|---|---|---|---|---|---|---|
| | Inadequate Laws and Regulations | Frequent Policy Adjustments | Imperfect Policy System | Vague Boundary of Property Rights | Imperfect Decision-Making System | Imperfect Accountability Mechanism | Unequal and Opaque Information | Complex Coordination Procedure | Lack of Expert Advice and Expertise | Imperfect (Operational) Management System | Inadequate Supervision | Unbalanced Distribution of Benefits | Single Financing Model | Inefficient Investment of Capital | Different Awareness of Stakeholders | Lack of Participation Willingness | Lack of Participation Policies | Problems of Social Equity |
| Government | | | | | | | | | | | | | | | | | | |
| Local and other administrative organizations | | | | | | | | | | | | | | | | | | |
| Consulting parties | | | | | | | | | | | | | | | | | | |
| Developers | | | | | | | | | | | | | | | | | | |
| Financial institutions | | | | | | | | | | | | | | | | | | |
| Owners | | | | | | | | | | | | | | | | | | |
| (General) Public | | | | | | | | | | | | | | | | | | |
| NGOs | | | | | | | | | | | | | | | | | | |
| Media | | | | | | | | | | | | | | | | | | |
| Designers | | | | | | | | | | | | | | | | | | |
| Contractors | | | | | | | | | | | | | | | | | | |
| Property party | | | | | | | | | | | | | | | | | | |
| Operating agencies | | | | | | | | | | | | | | | | | | |

**Table A2.** The relationship between stakeholders in urban renewal projects.

| | Rating of Relationship between Your Institution and Other Institutions | | | | | | | | | | Specific Strategies |
|---|---|---|---|---|---|---|---|---|---|---|---|
| | Before | | | | | After | | | | | |
| | 1 | 2 | 3 | 4 | 5 | 1 | 2 | 3 | 4 | 5 | |
| Government | | | | | | | | | | | 1. Providing policy guarantee: establishing special administrative agencies; formulating refined rules and regulations. |
| Local and other administrative organizations | | | | | | | | | | | 2. Improving the management system: establishing and improve relevant legal systems; through appropriate and reasonable regulatory measures. |
| Consulting parties | | | | | | | | | | | |
| Developers | | | | | | | | | | | |
| Financial institutions | | | | | | | | | | | 3. Establishing information exchange platforms. |
| Owners | | | | | | | | | | | 4. Providing financial support: finding other financial resources. |
| (General) Public | | | | | | | | | | | |
| NGOs | | | | | | | | | | | 5. Public participation: guaranteeing the rights of residents to participate. |
| Media | | | | | | | | | | | |
| Designers | | | | | | | | | | | |
| Contractors | | | | | | | | | | | |
| Property party | | | | | | | | | | | |
| Operating agencies | | | | | | | | | | | |

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
