# Peer review of "Overcoming the Collaboration Barriers among Stakeholders in Urban Renewal Based on a Two-Mode Social Network Analysis"

_land, doi:10.3390/land11101865_

Round 1
Reviewer 1 Report
The paper used a two-mode social network analysis to explore how to overcome the collaboration barriers among stakeholders in urban renewal in China. The methodology is clear, and the paper was structured fine. The following issues need to be addressed before publication.
1) the language of the paper needs to be polished. For instance, the title "overcoming collaboration barriers in urban renewal through stakeholders". Through stakeholders what? May it be Overcoming collaboration barriers among stakeholders in urban renewal?
2) the abstract should provide short background, method, and key findings of the study. "Finally, this study proposes strategies to promote collaboration between stakeholders from the five dimensions." My reading of this paper is five strategies were proposed, and their effectivenesses were tested.
3) the second paragraph needs to define the scope of urban renewal. What is urban renewal, and the key challenges in urban renewal?
4) the literature renew part on stakeholders in urban renewal needs to be clarified. The paper mentioned power and interest are two important characteristics of stakeholders. Only power was explored. The study needs to clarify what power means here. Also the second important thing the paper explored was collaboration. What does collaboration mean? Why is it important for urban renewal? What did the literature say about interaction and collaboration among stakeholders? These things are the theoretical foundations of the paper. They need to be clarified.
5) the result section proposed five strategies and the discussion section verified the strategies. Suggest to open a new section before the discussion to include the proposed strategies, further analysis on stakeholder collaborations in section 5 and verification of strategies.
6) some of the minor issues need to correct, e.g. change figure 3, 9 to table. add before(left), after(right) to figure 13.
Reviewer 2 Report
Thanks for the opportunity to review an interesting study about stakeholders and collaboration barriers in urban renewal. Although the study is interesting, there are some aspects that need to be considered in a revised version.
At the end of the paper, I have the impression that there was a concern in showing a lot of outputs (some unnecessary), but literature review, methods explanation or discussion need to be improved.
Introduction
A better link between knowledge gaps and your paper goals.
Literature Review
You can improve literature review regarding participation process and the main barriers. You can look for similar urbanization processes.
Methods
What is the study area? From where are experts? From a specific city, country? How you identify and contact experts?
A little confusion can exist when you refer to stakeholders. When you refer experts interviewed it is a single person. But when you refer to stakeholders identified (Table 2), if you don’t specify, I understand stakeholders as individual persons and not as stakeholders’ group. So, I recommend you put for example “Based on the above analysis, 13 stakeholder groups. were identified”. Correct this throughout the text.
On the step 2 the information should be organized. The questionnaires were distributed online and offline. Can you explain better this? Make the right relation to the information presented after that. You should refer the sample size on the methodology. How many experts did you contact initially and how many extra did you gain with snowballing? Can you explain the low coverage in some groups?
You should provide questionnaire as a supplement.
Results
Results should be more refined instead of presenting all the results available. Present those valuable ones please. There is some repetition in the forms of showing outputs. Table 5 and Figure 9 show similar results. Table 5 maybe is unnecessary output to show.
Attention on the Tables and Figures you should give necessary information, legend, to ensure they are self-explaining. Also, most of the captions are simple and need to be improved.
More comments in the paper pdf

Round 2
Reviewer 2 Report
Congratulations, you improved the paper and now I believe its ready to publish.